# High-Fidelity Operations on Silicon Donor Qubits Using Dynamical Decoupling Gates

**DOI:** 10.3390/e27080805

**Published:** 2025-07-28

**Authors:** Jing Cheng, Shihang Zhang, Banghong Guo, Huanwen Xie, Peihao Huang

**Affiliations:** 1Guangdong Provincial Key Laboratory of Nanophotonic Functional Materials and Devices, School of Optoelectonic Science and Engineering, South China Normal University, Guangzhou 510006, China; 2022022360@m.scnu.edu.cn; 2Guangdong Provincial Key Laboratory of Quantum Engineering and Quantum Materials, School of Optoelectonic Science and Engineering, South China Normal University, Guangzhou 510006, China; 3National Quantum Communication (Guangdong) Co., Ltd., Guangzhou 510700, China; vp_eng@nqctek.com; 4International Quantum Academy, Shenzhen 518048, China; 12331505@mail.sustech.edu.cn (S.Z.); huangpeihao@iqasz.cn (P.H.)

**Keywords:** dynamical decoupling gate, silicon-based phosphorus-doped system, quantum computing, spin qubits, fidelity

## Abstract

Dynamic decoupling (DD) can suppress decoherence caused by environmental noise, while in hybrid system it also hinders coherent manipulation between qubits. We realized the universal high-fidelity quantum gate set and the preparation of Bell states using dynamical decoupling gates (DD gates) in a silicon-based phosphorus-doped (Si:P) system, effectively resolving the contradiction between decoherence protection and manipulation of qubits. The simulation results show that the fidelity of the universal quantum gate set are all above 99%, and the fidelity of Bell state preparation is over 96%. This work realized the compatibility between coherent protection and high-fidelity manipulation of quantum states, provided a reliable theoretical support for high-fidelity quantum computing.

## 1. Introduction

The foundational framework for semiconductor spin qubits relies on electron spins and nuclear spins [1,2]. Semiconductor spin qubits have remarkable compatibility with the semiconductor industry and exhibit enormous scalability potential, featuring long coherence times and enabling high-fidelity electrical manipulation and readout. These attributes provide a robust foundation for integrating quantum computing systems with conventional semiconductor technology. Donor-based single-atom spin qubits provide a stable environment and longer spin relaxation times [3,4], with coherence times exceeding seconds in ^28^Si nuclear-spin-purified silicon crystals [5]. Key advancements in donor-based systems include fast-exchange two-qubit gates [6], electron-nuclear spin entanglement [7], and antimony-donor high-spin qubits [8]. Additionally, a high degree of freedom four-qubit register comprises three nuclear spins and one electron spin [9], as well as the creation and manipulation of Schrödinger cat states on antimony donors [10]. Previous studies have focused on qubit manipulation and coherence prolongation, while the compatibility between them in Si:P systems has not been considered.

The fidelities of quantum gates dictate the achievable depth and complexity of quantum algorithms, and error correction and fault tolerance can mitigate imperfections if single-qubit and two-qubit gate fidelities meet the error thresholds [11,12,13,14]. Decoherence remains a major barrier to the realization of scalable quantum technologies in solid-state systems. Quantum operations between qubits are inevitably perturbed by the solid-state environment; thus, the realization of high-fidelity gate operations is challenging. DD serves as a robust and practical tool for decoherence suppression, allowing each type of qubit to be decoupled at its own rate, ensuring uniform coherence protection [15]. Viola and Lloyd indicate that by continuously flipping qubits, the decoherence introduced by the environment can be effectively mitigated, and by making the intervals between pulses comparable to the environmental correlation time, decoherence will be greatly suppressed [16,17]. They further show that this noise suppression strategy can be combined with the dynamic evolution control of open quantum systems, providing a new theoretical framework for the coherence protection of quantum systems [18]. Subsequently, a series of optimization processes were performed on the pulse sequence. However, actual pulses will inevitably introduce errors in terms of intensity control and duration, and these errors will cause the DD scheme to fail to achieve the expected performance, while combining DD with composite pulses (CPs) can reduce the impact caused by errors, such as the well established CPMG-style spin echo [19], CDD [20], UDD [21], generalized UDD sequences [22], KDD [23], UR [24], crosstalk-robust DD [25], syncopated DD [26]. DD is widely applied in NV center [27,28,29], trapped ions [30,31], superconducting systems [32,33,34], etc., demonstrating its wide range of applications. To simulate the anti-noise property of DD pulses under real conditions, Franco et al. studied the robustness of various pulse sequences in a $\frac{1}{f}$ noise environment. Their research confirmed that the DD pulse sequence can effectively protect qubits from noise interference [35]. Although the combination of DD and quantum gate operations can isolate the coupling between target qubits and the environment, it generally suppresses interqubit interactions [36]. To solve the above problems, the DD gates are introduced, while it remains a critical challenge in solid state quantum information processing to achieve decoherence protection for qubits and universal high-fidelity quantum gate manipulation in Si:P system.

In this work, we design DD gates pulse sequences for a Si:P system, enabling high-fidelity universal quantum gate control. We analyze the dynamical evolution of quantum states while achieving single-qubit and two-qubit operations under coherence protection. Additionally, we study gate fidelities under quasi-static noise. Single-qubit gates $R_{y}\left(\frac{\pi }{2}\right)$ and $R_{x}\left(\frac{\pi }{4}\right)$ achieve 99.15% and 99.81%, respectively, while the two-qubit CNOT gate reaches 99.76% at 2000 Hz noise. Compared with unprotected gates, DD gates exhibit significant advantages. We also achieve Bell state preparation with a fidelity of 96.18% in a 1000 Hz noise environment, which fully demonstrates their strong robustness against environmental noise.

This article is structured as follows. In Section 2, we present the model and Hamiltonian of the Si:P system, which has a more stable environment and a longer relaxation time compared to other systems. After that, we show the principle of realizing the DD gates, setting the stage for understanding the circuit we designed in the next section. In Section 3, we use DD to slice the quantum state evolution into sets of shorter time periods. During the short evolution time, the electron spin is less affected by the environment. The coherence of the electron can be preserved as much as possible by applying DD pulse sequences in the short time. We further design quantum circuits to implement universal quantum gates and prepare Bell states. In Section 4, we analyze the enhancement of the fidelity of quantum gates manipulation by DD gates under quasi-static noise, and the fidelities of single-qubit and two-qubit quantum gates all exceed 99%, which meet the requirements of fault-tolerant quantum computation. In Section 5, we comparatively analyzed the single-qubit gate fidelity, two-qubit gate fidelity, and the number of decoupling pulses in the NV-center and Si:P system to further explore the advantages of our scheme. The comparison shows that DD gates can effectively improve the fidelity of universal quantum gate operation and Bell state preparation.

## 2. Theoretical Model

This section describes the model and quantum state operation principle of the Si:P system. To better understand the results in the next section, this section also explains the principle of the DD gates.

### 2.1. System Hamiltonian

The ^31^P donor in silicon behaves analogously to a hydrogen atom in solid-state system. The free electron and ^31^P nuclear in the Si:P system form a two-qubit system consisting of nuclear and electron spins [37]. In donor-based systems, the intrinsic confinement potential of doped atoms primarily localizes electrons, while external electrode potentials modulate spin qubit properties [38,39]. An external globally oscillating magnetic field is applied, usually within the radio frequency (RF) spectrum. The nuclear magnetic resonance (NMR) frequency is carefully chosen to resonate with the energy splitting of the donor nuclear spin state. This magnetic field induces a conditional nuclear spin flip on the electron spin state. When the frequency of the alternating magnetic field matches the Lamor frequency, the spin state transition is induced through the Rabi model. The amplitude and duration of the alternating magnetic field determine the spin flip angle. In the Si:P device, the voltage applied to the A-gate distorts the electron wave function, changing its overlap with the nuclear. This produces the ‘Stark shift’ [40]. The J-gate is used to attract electrons from neighboring donors, thereby controlling the overlap of their wave functions. By adjusting the coupling of each nuclear spin to neighboring nuclear spin and oscillatory fields, different operations can be performed on each nuclear spin simultaneously. The total Hamiltonian governing electron and nuclear spins in the Si:P system under external magnetic field can be expressed as [38](1)$$H_{0}=\gamma_{e}B_{0}S_{z}-\gamma_{n}B_{0}I_{z}+AS\cdot I,$$
where $B_{0}$ is the external magnetic field, $\gamma_{n}$ and $\gamma_{e}$ are the nuclear and electron gyromagnetic ratios, *S* and *I* represent electron and nuclear spin operators, with $S_{z}$ and $I_{z}$ being their z-components. The term $\gamma_{e}B_{0}S_{z}$ ($\gamma_{e}=27.97$ GHz/T) describes the electron Zeeman interaction, and $-\gamma_{n}B_{0}I_{z}$ (nuclear gyromagnetic ratio $\gamma_{n}=17.23$ MHz/T) [41] governs nuclear coupling. A≈117 MHz is the hyperfine interaction [42]. The matrix representation is(2)$$\left.H_{0}=\left(\begin{array}{cccc} \frac{z_{e}-z_{n}}{2}+\frac{A}{4} & 0 & 0 & 0\\ 0 & \frac{z_{e}+z_{n}}{2}-\frac{A}{4} & \frac{A}{2} & 0\\ 0 & \frac{A}{2} & \frac{-z_{e}-z_{n}}{2}-\frac{A}{4} & 0\\ 0 & 0 & 0 & \frac{-z_{e}+z_{n}}{2}+\frac{A}{4} \end{array}\right.\right),$$
where $z_{e}=\gamma_{e}B_{0}$ and $z_{n}=\gamma_{n}B_{0}$. The basis states follow the ordering |↓⇓〉, |↓⇑〉, |↑⇓〉, |↑⇑〉. |↑〉 and |↓〉 denote electron spin states are upward and downward with $S_{z}=\pm \frac{1}{2}$, |⇑〉 and |⇓〉 denote nuclear spin states are upward and downward with $I_{z}=\pm \frac{1}{2}$.

For the matrix, the diagonal elements represent the eigenenergy of the electron and nuclear spins. The ESR (NMR) frequencies of the electron (nuclear) are related to the nuclear spin (electron spin). The off-diagonal elements (arising from hyperfine interactions ${\hat{S}}_{x}\cdot {\hat{I}}_{x}+{\hat{S}}_{y}\cdot {\hat{I}}_{y}$) represent the quantum coupling strength between different ground states, and this coupling drives Rabi oscillations, which is the basis for manipulating spin states. In addition, off-diagonal elements can lead to weak mixing of spin states, causing energy level shifts, and the weak mixing is usually neglected. The electron spin transition frequencies are $\nu_{e1}=E_{↑⇓}-E_{↓⇓}=-z_{e}-\frac{A}{2}$ (nuclear spin |⇓〉) and $\nu_{e2}=E_{↑⇑}-E_{↓⇑}=-z_{e}+\frac{A}{2}$ (nuclear spin |⇑〉). The nuclear transition frequencies are $\nu_{n1}=E_{↓⇓}-E_{↓⇑}=-z_{n}+\frac{A}{2}$ (electron |↓〉) and $\nu_{n2}=E_{↑⇑}-E_{↑⇓}=z_{n}+\frac{A}{2}$ (electron |↑〉). The energy level structure of the electron—^31^P nuclear spin system is shown in Figure 1.

The implementation of electron spin resonance (ESR) and nuclear magnetic resonance (NMR) can be achieved by imposing an alternating magnetic field on the antenna. The manipulation of electron and nuclear spins is accomplished through the selection of specific frequencies for ESR and NMR. Microwave (MW) pulses in the GHz range can regulate electron spin transitions, while RF pulses in the MHz range can drive nuclear spins through resonance.

When the qubit is in a static magnetic field $B_{0}$ along the z-axis, it is also affected by a driving magnetic field $B_{1}$ along the y-axis. The Hamiltonian contains the y-component operators of electron spin and nuclear spin(3)$$H_{1}=B_{1}\left(t\right)\left[\gamma_{e}\left({\hat{S}}_{y}⊗I\right)+\gamma_{n}\left(I⊗{\hat{I}}_{y}\right)\right],$$
where ${\hat{S}}_{y}⊗I$ drives electron spins via their y-component operator ${\hat{S}}_{y}$, while $I⊗{\hat{I}}_{y}$ drives nuclear spins through ${\hat{I}}_{y}$. The matrix form of $H_{1}$ is(4)$$H_{1}=B_{1}\left(\begin{array}{cccc} 0 & -i\gamma_{n} & -i\gamma_{e} & 0\\ i\gamma_{n} & 0 & 0 & -i\gamma_{e}\\ i\gamma_{e} & 0 & 0 & -i\gamma_{n}\\ 0 & i\gamma_{e} & i\gamma_{n} & 0 \end{array}\right),$$

By applying corresponding RF pulses and MW pulses based on this Hamiltonian, qubits can be manipulated. This paper uses the numerical solution to solve the time evolution of the Schrodinger equation. Specifically, by discretizing the time and solving the differential equation system corresponding to the Schrodinger equation, the quantum state evolution under the time-dependent Hamiltonian is calculated for each time t. The evolved quantum state is obtained through numerical solutions, and finally, the final state is obtained, and the fidelity of the quantum state is calculated. At the same time, the influence of noise can be simulated by introducing the environmental coupling term. This better aligns the simulation results with reality.

### 2.2. DD Gate Implementation Principle

This work establishes an operational framework for DD gates in the Si:P system. Based on average Hamiltonian theory [43], the time-dependent evolving Hamiltonian in a long time can be cut into the sum of time-independent Hamiltonian in a short time, which simplifies the complex Hamiltonian into a more tractable form and makes the regulation of quantum states more operable. Precise manipulation of the quantum states was performed by coupling and driving the electron spin and nuclear spin of ^31^P.

We use the simplest XY-4 DD sequences as an example to illustrate its specific implementation process. Figure 2 shows a schematic diagram of the pulse sequence for implementing the DD gates. The quantum gate can be achieved by hyperfine interactions, and the free evolution around z-axis can be expressed as $\tau =\frac{\pi }{A}$. During the evolution time τ between DD pulses, an appropriate split-rotating gate operation is applied. When conducting theoretical analysis, we assume that each evolution time interval is sufficiently short and satisfies the time-independent evolution condition, the gate operation over the total duration $\tau_{c}$ can be decomposed into a sequence of unitary transformations applied during the intervals between DD pulses. In Figure 2, $\tau_{c}=4\tau$. Through this method, quantum gate manipulation can be completed while protecting the system from environmental noise interference [28].

During nuclear spin manipulation, when applying pulse sequences to implement single-qubit gates, it is necessary to ensure that the control subspaces of the RF pulses between each DD pulses are consistent. All RF pulses are selected as either $\nu_{n1}$ (flipping the nuclear spin when the electron spin is in the |↓〉 state) or $\nu_{n2}$ (flipping the nuclear spin when the electron spin is in the |↑〉 state). Assuming that the rotation angle within each time interval $\tau_{n}$ is $\theta_{n}$, where *n* is the number of τ, we choose Aτ=2nπ. If the electron is in state |↑〉 (|↓〉) during τ, the nuclear spin rotates 180° about the z-axis. During the time interval 2τ, the nuclear spin undergoes a 360° rotation that has zero net effect. Meanwhile, the spin rotation angle of the subspace of electron |↓〉 (|↑〉) is $\theta_{n}$. Since the electron subspace is constantly switched, the applied driving frequency remains constant. The total rotation angle is $\theta =\frac{n}{2}\theta_{n}$.

For two-qubit quantum gate operations, the control subspace of the RF pulses must be switched between DD pulses. First, a pulse with frequency $\nu_{n1}$ is applied (flipping the nuclear spin when the electron spin is in the |↓〉 state), followed by a DD pulse sequence to flip the electron state by an angle of π. After this operation, the subspaces of the |↑〉 and |↓〉 states are swapped. If it is necessary to perform a nuclear spin flip operation of angle $\theta_{n}$ within the same electron subspace, the frequency of the RF pulse must be switched to $\nu_{n2}$ (flipping the nuclear spin when the electron spin is in the |↑〉 state) after applying the DD pulse sequences, and the DD pulse sequences must be applied again. By cyclically switching subspace and driving frequencies, the two-qubit gate operation is achieved, with the total rotation angle $\theta =n\theta_{n}$.

We have clarified the principle of using DD gates to realize qubit coherence protection and precise control. The specific implementation methods and corresponding circuit diagrams of the single-qubit quantum gate $R_{x}\left(\theta \right)$ and the two-qubit quantum gate $CR_{x}\left(\theta \right)$ with any rotation angle θ are detailed in Appendix A. Next, we will implement high-fidelity universal quantum gate operations using DD gates in the Si:P system. The aim of this work is to improve the coherence time of the system and realize high-fidelity quantum state control under environmental noise.

## 3. Universal Quantum Gate Realization

This section demonstrates the implementation of universal quantum gates (Hadamard gate, Phase gate, T gate, and CNOT gate) via the DD protocol. By embedding spin-echo sequences into the gate operation, quantum state manipulation is realized while suppressing the error [44].

### 3.1. Universal Single-Qubit Gates Realized via DD Gates

Single-qubit gates in a discrete universal gate set typically include the Hadamard gate (H gate), Phase gate (S gate), and T gate ($\frac{\pi }{4}$ gate). In the implementation of quantum state manipulation, the H gate is typically replaced by an $R_{y}\left(\frac{\pi }{2}\right)$ rotation. The Phase gate applies a $\frac{\pi }{2}$ phase rotation to the |↓〉 (|⇓〉) state, while keeping the probability amplitude unchanged. The T gate performs $\frac{\pi }{4}$ phase rotation on the |↓〉 (|⇓〉) state, equivalent to the operation of $R_{z}\left(\frac{\pi }{4}\right)$. The S gate corresponds to the $R_{z}\left(\frac{\pi }{2}\right)$ operation [44]. The $R_{z}\left(\theta \right)$ rotation can be synthesized by the following composite operation(5)$$R_{z}\left(\theta \right)=R_{y}\left(\frac{\pi }{2}\right)R_{x}\left(-\theta \right)R_{y}\left(-\frac{\pi }{2}\right).$$

The above equation indicates that the $R_{z}\left(\frac{\pi }{2}\right)$ gate (S gate) and $R_{z}\left(\frac{\pi }{4}\right)$ gate are realizable via $R_{y}\left(\frac{\pi }{2}\right)R_{x}\left(-\frac{\pi }{2}\right)R_{y}\left(-\frac{\pi }{2}\right)$. Thus, DD gates supporting $R_{y}\left(\frac{\pi }{2}\right)$ and $R_{x}\left(\frac{\pi }{4}\right)$ operations enable universal single-qubit control.

Figure 3 shows the quantum circuits constructed based on DD gates. The circuits are used to implement the $R_{y}\left(\frac{\pi }{2}\right)$ (Figure 3a) and $R_{x}\left(\frac{\pi }{4}\right)$ (Figure 3b) operations. Single-qubit quantum gates use fixed frequency RF pulses, where DD pulses manipulate nuclear spins by switching electronic subspaces. Since $\theta =\frac{n}{2}\theta_{n}$, when implementing the $R_{y}\left(\frac{\pi }{2}\right)$ and $R_{x}\left(\frac{\pi }{4}\right)$ operations using the XY-4 DD sequences, the rotation angles within the τ interval should be set to $\frac{\pi }{4}$ and $\frac{\pi }{8}$, respectively. In this paper, the total DD gate operation times of the single-qubit $R_{y}\left(\frac{\pi }{2}\right)$ and $R_{x}\left(\frac{\pi }{4}\right)$ gates are 0.103 ms and 0.0526 ms, respectively. The total evolution time of the quantum gates implemented using the DD sequences are determined by the DD pulses interval. In addition, the quasi-static noise time is long enough, ensuring that all gate operations are under noise conditions and guaranteeing the authenticity of the simulation results. Figure 3a’,b’ shows the dynamical evolution process of single-qubit quantum gates $R_{y}\left(\frac{\pi }{2}\right)$ and $R_{x}\left(\frac{\pi }{4}\right)$ using DD gates. We will use Figure 3a’ as an example to illustrate the $R_{y}\left(\frac{\pi }{2}\right)$ process.

Initial state of the system is selected as |↑⇓〉.The drive frequency $\nu_{n1}$ (the drive frequency of the nuclear spin when the electron spin is |↓〉) cannot drive the nuclear spin.X pulse is applied, causing the electron spin to rotate by π around the x-axis, changing the electron spin from |↑〉 to |↓〉, resulting in the quantum state |↓⇓〉. This process corresponds to the evolution of the blue circular curve (|↓⇓〉) and the green triangle curve (|↑⇓〉).We apply another RF pulse with a frequency of $\nu_{n1}$ to drive the nuclear spin to rotate around the y-axis by $\frac{\pi }{4}$, placing the nuclear spin in a superposition state of |⇑〉 and |⇓〉.Y pulse is applied to drive the electron spin to rotate around the y-axis by an angle of π, returning the electron spin to the |↑〉 state.

The DD pulse sequences can periodically switch the electronic spin subspace, realizing the selective driving of nuclear spins within τ. Repeating the above operations, applying an even number of DD pulses can realize non-selective driving of nuclear spins. The above example demonstrates the single-qubit gate manipulation process using the DD gates to implement the $R_{y}\left(\frac{\pi }{2}\right)$ gate. It should be noted that in actual quantum manipulation, when using single-qubit rotation gates or combined rotation gates to perform operations, global phase factors may be introduced. Since global phase factors are unobservable, and they are typically neglected.

### 3.2. Universal Two-Qubit Gates Realized via DD Gates

In the set of discrete universal quantum gates, the two-qubit quantum gate is the CNOT gate. Specifically, the CNOT gate can be implemented using the control rotation gate $CR_{x}\left(\pi \right)$ or $CR_{y}\left(\pi \right)$. We use the XY-4 pulse sequences to implement two-qubit gate manipulation and the operation time of CRy(π) gate is 0.169 ms. As shown in Figure 4a, nine pulse sequences are used to implement the CNOT gate. Next, we will explain the implementation process of the CNOT gate in detail.

To ensure that the results effectively show the CNOT gate operation rather than the NOT gate operation, we select the initial state as $\frac{1}{\sqrt{2}}\left(|↑⇓\rangle +|↓⇓\rangle \right)$. The specific implementation process of two-qubit gate is as follows.

Applying a selective RF pulse to the nuclear spin. When the electron is in the |↓〉 state, the nuclear spin is rotated by $\frac{\pi }{8}$, leaving the electron spin in a superposition state while the nuclear spin remains in the eigenstate.Y pulse is applied to rotate the electron spin by π around the y-axis, causing the electron subspace to swap between |↑〉 and |↓〉.An RF pulse with the frequency of $\nu_{n1}$ is applied to flip the nuclear spin by $\frac{\pi }{4}$ when the electron spin is in the |↑〉 state.Repeat the above process until the target state $\frac{1}{\sqrt{2}}\left(|↑⇓\rangle -|↓⇑\rangle \right)$ is obtained.

The dynamical evolution of the controlled rotation operation is shown in Figure 4a’. The initial superposition state (blue circular curve and green triangle curve) evolves under the action of the $\nu_{n1}$ RF pulse, with the probability of the |↓⇓〉 state (blue circular curve) decreasing and the probability of the |↓⇑〉 state (orange square curve) increasing, while the probability of the |↑〉 subspace (green triangle curve/red diamond curve) remains unchanged. This confirms that the circuit effectively achieves selective control of the nuclear spin in the electron |↓〉 state. Based on the dynamical evolution of the quantum state and the visualization of the final state density matrix, it can be observed that the initial state $\frac{1}{\sqrt{2}}\left(|↓⇓\rangle +|↑⇑\rangle \right)$ evolves into the final state $\frac{1}{\sqrt{2}}\left(|↑⇓\rangle -|↓⇑\rangle \right)$, effectively implementing a two-qubit CNOT gate operation. Ideally, the DD gates is able to efficiently maintain the coherence of the quantum system while realizing the fault-tolerant manipulation of nuclear spin rotation.

### 3.3. Bell State Preparation Enabled by DD Gates

The preparation of high-fidelity Bell states in different physical systems is one of the important goals in realizing fault-tolerant quantum computing. It exhibits nonlocal quantum correlations and is the basis for realizing scalable quantum computation [45,46]. The quantum entangled state also shows a significant application value in the field of quantum communication. Based on the Greenberger–Horne–Zeilinger (GHZ) entangled state, a measurement device-independent quantum key distribution (MDI-QKD) scheme involving multiple users can be constructed, which exhibits remarkable flexibility in terms of system architecture and application scenarios [47]. In practical physical systems, the presence of environmental noise significantly reduces the coherence of qubits, which in turn leads to the decoherence of entangled states between qubits. To address this key challenge, this study proposes a DD gate-based method for the preparation of Bell states. By introducing DD sequences in the preparation process, the impact of environmental noise on the coherence of qubits can be effectively suppressed. Meanwhile, the driving pulse sequences are embedded within the time gap of the DD sequence to realize the efficient preparation of Bell states. In the following we will specify the preparation of the Bell state.

We choose |↑⇓〉 as the initial state, and first perform the $R_{y}\left(\frac{\pi }{2}\right)$ operation on the electron to prepare electron spin in the superposition state of |↑〉+|↓〉. This is the difference from the implementation of the CNOT gate. Subsequently, a selective RF pulse is applied to the nuclear spin. Since the electron spin decoherence time is on the order of microseconds (μs) and the nuclear spin decoherence time is on the order of milliseconds (ms), there is a significant difference between them. In order to suppress the decoherence effect of electron spins during nuclear spin operation, DD gates are used to realize the CNOT gate. When the electron is in the |↑〉 state, the nuclear spin is driven to rotate −π around the y-axis, and the Bell state $\frac{1}{\sqrt{2}}\left(|↑⇑\rangle +|↓⇓\rangle \right)$ is finally prepared.

Figure 4b’ indicates that the DD gates successfully prepared the Bell state. A superposition of |↑⇑〉 (red diamond curve) and |↓⇓〉 (blue circular curve) was generated. Next we study the performance of the operation in the presence of the environmental noise.

## 4. Fidelity Analysis

We analyzed the fidelity of qubit manipulation under environment noise interference for DD gates versus conventional pulse-driven techniques. Special attention is paid to quasi-static noise. This type of noise is characterized by slow, low-frequency fluctuations caused by environmental variations or internal parameter shifts. It introduces random phase changes in the qubits. These variations will lead shifts in the population of the quantum state and reduce the fidelity of the overall system [48]. To approximate such noise, static Pauli-Z noise was introduced in the simulations. The overall noise intensity is set by $\delta_{e}$, which also represents the noise intensity for the electron spin. The nuclear spin noise strength is determined by the coefficient $\eta =\delta_{n}/\delta_{e}$, where $\delta_{n}$ is the noise intensity for the nuclear spin. To characterize the behavior dominated by dephasing, the noise intensities are incorporated into the Hamiltonian as qubit frequency detunings(6)$$H_{\delta }=\delta_{e}S_{z}+\delta_{n}I_{z}.$$

In order to systematically compare the adaptability of the two methods to this noise source, we measured the fidelity at different quasi-static noise intensities. The results show that the DD gates can effectively suppress the effect of environmental noise in the quantum system. In addition, it has significant advantages in maintaining system coherence and improving the fidelity of qubit manipulation. The comparative analysis of the electron spin qubit fidelity in Figure 5 clearly shows the noise tolerance of the DD gates over the entire test noise range.

In the DD gates theory simulation of the Si:P system, the noise intensity is controlled in the range of 0–2000 Hz. We choose the $\frac{1}{\sqrt{2}}\left(|↑⇓\rangle +|↓⇓\rangle \right)$ state as the initial state. The results show that the fidelity of the DD gates is lower than that of a single NMR operation with no noise (Figure 5a). This mainly stems from the systematic error of quantum gates operation. That is, the DD gate operation is realized by a combination of multiple gates, whose cumulative error is higher than that of a single NMR pulse, resulting in lower fidelity under ideal conditions.

Figure 5a shows single-qubit gates for $R_{y}\left(\frac{\pi }{2}\right)$ and $R_{x}\left(\frac{\pi }{4}\right)$. The comparison shows that the fidelity of the single NMR operation is lower than the DD gates. The electronic fidelity tends to decrease with an increase in noise intensity. This is particularly evident for long duration operations with larger rotation angles. Figure 5b compares the fidelity of the CNOT gate. A single NMR operation is indicated by the blue square solid line. The electronic fidelity decreases rapidly because no protection is applied to the electron spin during the nuclear spin evolution. The DD gates effectively maintains the electronic coherence by embedding a DD pulse sequences in the nuclear spin operation and continuously flipping the electron spin. Due to the longer nuclear spin coherence time and lower sensitivity to noise, the nuclear spin has higher fidelity under a noisy environment. It is shown that the DD gates can realize high-fidelity universal quantum operations in the Si:P system. Compared with single NMR nuclear spin control, the DD gates can improve the overall system fidelity by a factor of about 10. In addition, the fidelity of the Bell state preparation (green square curve) is also superior to that of the unprotected conventional method (blue circular curve), as shown in Figure 5c. The preparation fidelity is as high as 98.61%, on average, under noise.

Figure 5a’,b’ shows the real part of the density of final state obtained after single-qubit gate as well as two-qubit operations on the initial quantum state $\frac{1}{\sqrt{2}}\left(|↑⇑\rangle +|↓⇓\rangle \right)$ in the noisy environment. The single-qubit gate operations are $R_{y}\left(\frac{\pi }{2}\right)$ and $R_{x}\left(\frac{\pi }{4}\right)$. The two-qubit gate operation is $CR_{y}\left(\pi \right)$. Through analysis of the data in the figures, it is evident that the population of the final states are in good agreement with theoretical expectations, indicating that the quantum states are less perturbed by environmental noise. Figure 5c’ presents the real part of the density matrix of the Bell states prepared under the noise environment, where the population of |↑⇑〉 and |↑⇓〉 are 0.4998 and 0.4992, respectively. It is further found that the average fidelity of the system can reach 96.18% when the noise intensity is 1000 Hz. These simulation results demonstrate that DD gates can effectively enhance the system fidelity of discrete universal quantum gate operations and Bell state preparation, providing important support for fault-tolerant quantum computing.

## 5. Discussion and Conclusions

Previous research on DD gates has been primarily limited to NV-center system [49,50], with no systematic exploration conducted in Si:P system and no comprehensive validation of the feasibility [15,28] of universal gate. In this article, DD gates are employed to construct single-qubit and two-qubit quantum circuits in the Si:P system, enabling the effective implementation of a discrete universal quantum gate set. In addition, this work also realized the Bell state preparation by DD gates.

We compared the fidelity of single-qubit quantum gates, the fidelity of two-qubit quantum gates, and the fidelity of Bell state preparation to highlight the advantages of this scheme. To ensure the comparability of the results, for the experimental schemes mentioned in Table 1, we first calibrated different noise intensities without applying DD gates. Subsequently, the noise intensity consistent with the experimental results in the comparison schemes is selected and the DD gates are applied for comparison. The number of decoupling pulses is 4 which is shown in Table 1. In the constructed universal quantum gate set in the Si:P system, the average fidelity of single-qubit gates reaches 99.8%, and that of two-qubit gate reaches 99% in a small bath (500 Hz), which is a significant increase in fidelity compared with that of single-qubit gates (traditional NMR control 98.2%, 98.4% [51], 99.15% [52] and 98% [53]) and two-qubit gates (traditional NMR control 97.5%, 96.4% [51] and 98% [52]). In addition, the average fidelity of Bell state preparation realized based on the DD gates is 94.61%, which is much higher than that of traditional NMR control 85%, 89% [52]. This paper adopts analytical methods, which are significantly different from the existing high-fidelity control technologies (such as GRAPE and DRAG). GRAPE relies on numerical methods, and the waveforms are relatively complex. It is difficult to generate complex waveforms under the condition of short electron coherence time [54]. DRAG mainly optimizes for high-level leakage [55]. In contrast, this paper focuses on the problem that the significant differences in the manipulation time scale between electron spin and nuclear spin make universal control difficult to achieve, the DD gate is designed, and this scheme does not require special optimization for specific systems. Although we have theoretically demonstrated the feasibility of realizing high-fidelity gates using the DD gate, there are still many challenges in implementing the DD gate in the Si:P system. The deviation of qubit rotation angle caused by the inaccurate DD pulse time is an important factor (As shown in Appendix B). In addition, the gate manipulation process generates excessive dissipation and heat [56,57,58,59,60]. The increase in temperature will affect the stability of the system and thus reduce the overall fidelity of the system.

Based on simulation results and evolution analyses, we know that DD gates can effectively enable the manipulation of universal quantum gate sets and the preparation of Bell states in Si:P system. In the presence of noise, the fidelities of single-qubit gates, two-qubit gates, and Bell state preparation are improved by 0.65–1.8%, 1–2.6%, and 5.61–9.61%, respectively. Compared with the above schemes, the proposed scheme has more advantages for the following reasons: First, it can realize the coordination of gate operation and noise suppression. Second, with the help of DD gate, the time interval of manipulation can be flexibly adjusted to ensure that the noise is refocused before accumulation, which significantly enhances the ability to suppress quantum state decoherence. Third, considering that the noise in the silicon-based system is mainly low-frequency noise, the characteristics of the DD gate can perfectly adapt to the system. Overall, this scheme realized the coherent protection of quantum states and high-fidelity quantum manipulation in Si:P systems and high-fidelity operations with few decoupling pulses. These results provide theoretical support for the realization of scalable fault-tolerant quantum computing. In addition, the multiple ^31^P nuclear spins in Si:P can be coupled through electron spins. Using DD sequences synchronously for protection could be considered. This paper mainly considers the implementation of high-fidelity gates in single-electron and single-nuclear systems. In principle, the DD gate can be extended to the case of a multi-donor system, and this will be investigated in more detail in the future. The existence of crosstalk and electrical noise in multinuclear systems is an important challenge in achieving high-fidelity quantum gates, and they represent the problems that we need to solve in the future. 

## Figures and Tables

**Figure 1 entropy-27-00805-f001:**
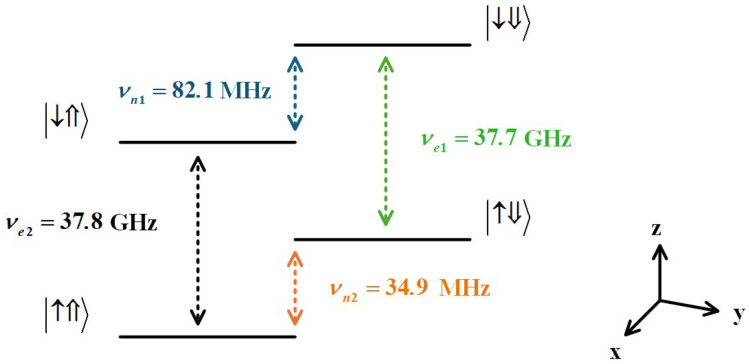
Energy level structure of electron—^31^P nuclear spin system. The external magnetic field $B_{0}=1.35$ T, along the z direction, hyperfine A=117 MHz, electron gyromagnetic ratio $\gamma_{e}=27.97$ GHz/T, nuclear gyromagnetic ratio $\gamma_{n}=17.23$ MHz/T. Under the above conditions, $\nu_{e1}$ = 37.7 GHz (green) and $\nu_{e2}$ = 37.8 GHz (black) represent the transition frequencies of the electron spin while the nuclear spin is in the |⇓〉 state and |⇑〉. $\nu_{n1}$ = 82.1 MHz (blue) and $\nu_{n2}$ = 34.9 MHz (orange) represent the transition frequencies of the nuclear spin when the electron spin is in the |↓〉 and |↑〉, respectively.

**Figure 2 entropy-27-00805-f002:**
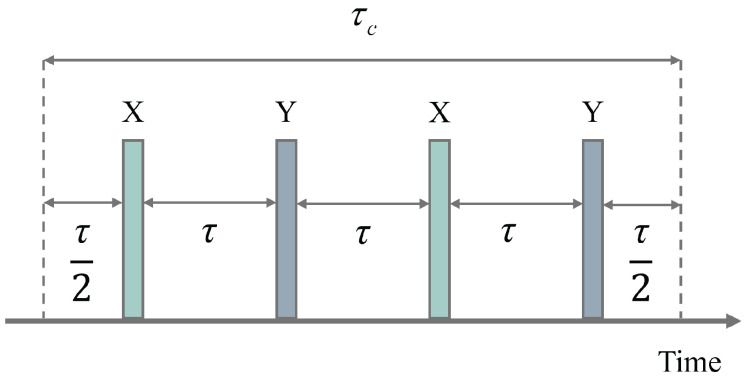
Schematic diagram of the protection gate operation pulse realized by XY-4 DD sequences. X and Y denote that the DD pulses drive the quantum state to rotate by π around the x-axis or y-axis, respectively. The X pulses and Y pulses are respectively represented by green and blue bars. Continuous electron decoupling pulses are applied on the orthogonal axes to reduce the accumulation of pulse errors. $\tau_{c}$ represents the evolution time of the entire system, with $\tau_{c}=4\tau$. $\tau_{c}$ satisfied the condition Aτ=2nπ, and the total duration of the pulse is an integer multiple of the driving period to avoid the net Stark shift of the electron spin caused by the nuclear pulse. During the application of MW pulses, the nuclear drive is turned off to prevent MW pulse detuning caused by the RF pulses.

**Figure 3 entropy-27-00805-f003:**
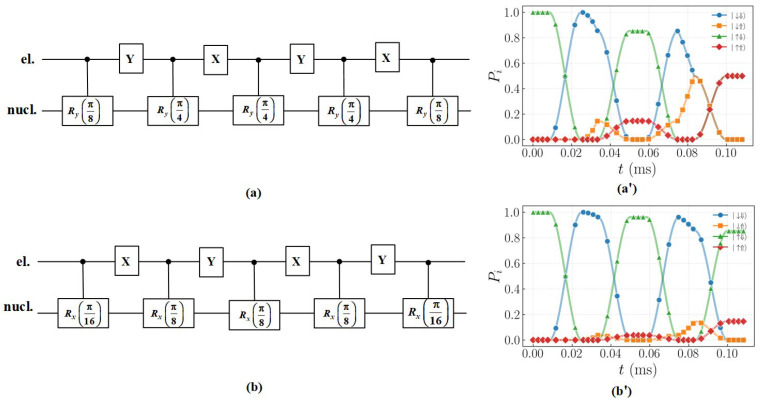
Single-qubit quantum gate circuit. (**a**) $R_{y}\left(\frac{\pi }{2}\right)$ quantum gate operation. (**b**) $R_{x}\left(\frac{\pi }{4}\right)$ quantum gate operation. (**a’**) Dynamical evolution diagram of the single-qubit $R_{y}\left(\frac{\pi }{2}\right)$ gate implemented by the DD gates. (**b’**) Dynamical evolution diagram of the single-qubit $R_{x}\left(\frac{\pi }{4}\right)$ gate implemented by the DD gates. By applying a single frequency $\nu_{n1}$ over τ, the nuclear spin is flipped when the electron spin is in the |↓〉 state. Subsequently, subspace switching via DD pulses enables indirect manipulation within the two subspaces of the electron, ultimately achieving single-qubit quantum gate operations. Starting from the initial state |↑⇓〉, after passing through the $R_{y}\left(\frac{\pi }{2}\right)$ and $R_{x}\left(\frac{\pi }{4}\right)$ gate, the final state are the superposition state of |↑⇑〉 and |↑⇓〉.

**Figure 4 entropy-27-00805-f004:**
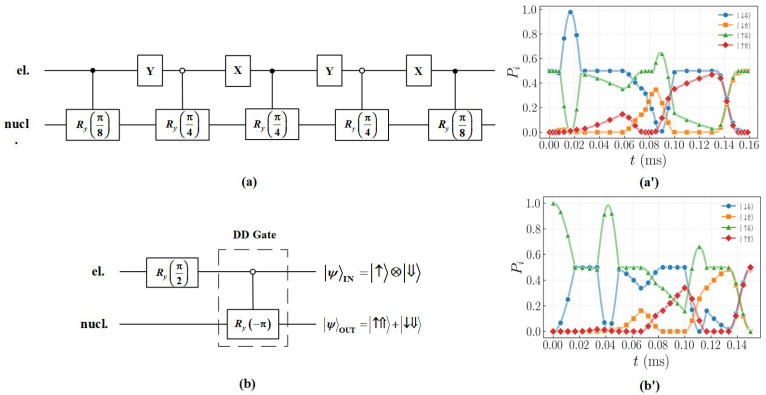
Two-qubit quantum gate circuit and dynamical evolution. (**a**) Quantum circuit of the CNOT gate. This circuit implements an operation that rotates the nuclear spin around the y-axis by an angle of π when the electron spin is in the |⇓〉 state. To achieve this two-qubit gate, the frequency of the RF pulses must be continuously adjusted to counteract the subspace switching effect of the DD pulses on the electron spin, thereby enabling the driving of the nuclear spin within specific electron subspace. (**a’**) Dynamical evolution of the CNOT gate. This diagram illustrates the dynamical evolution of the nuclear spin being driven to rotate around the y-axis by an angle of π via RF pulse when the electron spin is in the |↓〉 state. The target state $\frac{1}{\sqrt{2}}\left(|↑⇓\rangle -|↓⇑\rangle \right)$ is obtained after the CNOT gate of the initial state $\frac{1}{\sqrt{2}}\left(|↑⇓\rangle -|↓⇑\rangle \right)$. The solid lines depict the evolution processes of the corresponding quantum states. (**b**) Quantum circuit diagram for preparing a Bell state using DD gates. (**b’**) Dynamical evolution diagram of Bell state preparation. Starting from the initial state |↑⇓〉, the Bell state $\frac{1}{\sqrt{2}}\left(|↑⇑\rangle +|↓⇓\rangle \right)$ is prepared by first applying an $R_{y}\left(\frac{\pi }{2}\right)$ rotation to place the electron spin in a superposition state, followed by a controlled $CR_{y}\left(-\pi \right)$ operation on the nuclear spin when the electron spin is in the |↑〉 state.

**Figure 5 entropy-27-00805-f005:**
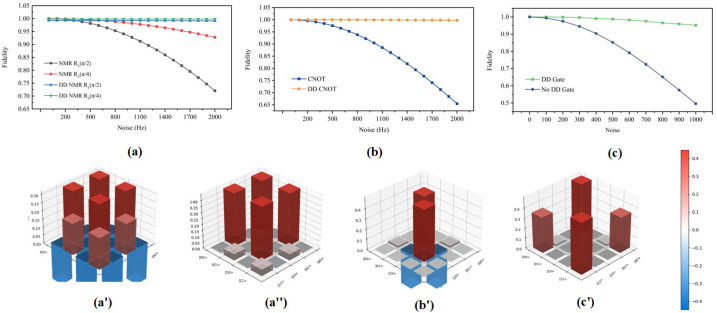
Fidelity comparison of universal quantum gate sets and Bell state. (**a**) Fidelity comparison for single-qubit gates $R_{y}\left(\frac{\pi }{2}\right)$ and $R_{x}\left(\frac{\pi }{4}\right)$. (**b**) Fidelity comparison for the controlled rotation gate $CR_{y}\left(\pi \right)$. This diagram also illustrates the dynamical evolution of the nuclear spin being driven to rotate around the y-axis by an angle of π via RF pulses when the electron spin is in the |↓〉 state. To implement the two-qubit gate, the frequency of the selective pulses must be continuously adjusted to drive the nuclear spin within specific electron subspaces. (**c**) Fidelity comparison for Bell state preparation. The green square line represents the fidelity of Bell state preparation using DD gates. The blue circular line represents the fidelity of Bell state preparation without applying DD protection. The bottom line (**a’**,**a”**,**b’**) shows the real parts of the density matrices for the final states of discrete universal quantum gate sets under noisy conditions. From left to right, we depict the real parts of the density matrices after applying $R_{y}\left(\frac{\pi }{2}\right)$, $R_{x}\left(\frac{\pi }{4}\right)$, and $CR_{y}\left(\pi \right)$ operations to the initial state $\frac{1}{\sqrt{2}}\left(|↑⇓\rangle +|↓⇓\rangle \right)$. (**c’**) Real part of the density matrix for the Bell state generated by applying $R_{y}\left(\frac{\pi }{2}\right)$ on the electron spin followed by $R_{y}\left(-\pi \right)$ on the nuclear spin when the electron is in the |↑〉 state, starting from the initial state |↑⇓〉 under noisy conditions. Red denotes positive values, blue denotes negative values, and the color intensity represents the magnitude of the corresponding element.

**Table 1 entropy-27-00805-t001:** Comparison of qubit manipulation fidelity.

Parameter	Single Qubit Fidelity	Tow Qubit Fidelity	Bell State Fidelity
Our System	99.8%	99%	94.61%
NMR	98.2%	97.5%	85%
R Savytskyy [51]	98.4%	96.4%	–
W. Huang [52]	99.15%	98%	89%
Jarryd J. Pla [53]	98%	–	–

## Data Availability

Data are contained within the article.

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
