# Peer review of "High-Fidelity Operations on Silicon Donor Qubits Using Dynamical Decoupling Gates"

_entropy, 2025, doi:10.3390/e27080805_

Round 1
Reviewer 1 Report
Comments and Suggestions for Authors
This paper proposes a research method based on theory and simulation. The universal quantum gate set is realized by using the dynamic decoupling (DD) pulse sequences in the silicon-based phosphorus-doped system (Si:P). The authors demonstrate through numerical simulation that the DD gate can protect electron coherence and achieve high-fidelity quantum gate operations. The gate fidelity of single-qubit and tow-qubit gates exceed 99$\%$, and the fidelity of Bell state preparation under quasi-static noise exceeds 96$\%$.
This work solves the challenge of the coherent protection and high-fidelity gate operations of quantum states in solid-state hybrid systems.
The manuscript is well structured and the method is reliable, and it has some scientific significance.
I suggest that this paper can be published after a little revision.
These results provide a theoretical reference for implementing universal quantum gate sets with dynamic decoupling gates. It applies to solid-state hybrid systems, especially Si:P systems.
The fidelity is competitive. If it can be confirmed by experiments, the application prospect will be more extensive. Authors are asked to consider the following questions:
1 This paper discusses the effect of quasi-static noise on the fidelity of quantum gates, but does not mention the gate duration or compare it with the gate duration in noise. Can the authors show the time scale of the gate operation and compare it with the duration of the gate under noise?
2. Although this paper is based on theoretical simulation, it would be more valuable if the feasibility of implementing DD gates in Si:P devices could be briefly discussed. What challenges will be faced in achieving the required DD pulse sequences and pulse control accuracy in the experiment.
3. It is assumed in paper that the pulse is ideal. In fact, when applying the pulse sequences, errors will be introduced: the rotation angle may be too large or too small, and the rotation axis may also be offset. These are all important factors that bring errors. For these non-ideal cases, how effective is the DD gate sequences in enhancing the robustness of the system.
4. This manuscript focuses on single-qubit and tow-qubit operations. Could the authors discuss whether this approach has the same effect on multi-qubit systems?
5. The authors compared their method with traditional NMR. It would be more useful to briefly compare DD gates with high-fidelity control techniques such as GRAPE and DRAG.
6. If this paper involves specific software packages or numerical solvers, please supplement the references of the simulation tools or methods used.
Author Response
For research article
Response to Reviewer 1 Comments
|
||
1. Summary |
|
|
Thank you very much for reviewing this manuscript in your busy schedule. Please find the detailed reply below. The corresponding revised parts are also highlighted in the resubmitted documents.
|
||
2. Questions for General Evaluation |
Reviewer’s Evaluation |
Response and Revisions |
Does the introduction provide sufficient background and include all relevant references? |
Yes |
|
Are all the cited references relevant to the research? |
Yes |
|
Is the research design appropriate? |
Yes |
|
Are the methods adequately described? |
Can be improved |
|
Are the results clearly presented? |
Yes |
|
Are the conclusions supported by the results?
|
Yes |
|
3. Point-by-point response to Comments and Suggestions for Authors |
||
Comments 1: This paper discusses the effect of quasi-static noise on the fidelity of quantum gates but does not mention the gate duration or compare it with the gate duration in noise. Can the authors show the time scale of the gate operation and compare it with the duration of the gate under noise? |
||
Response 1: Thank you for pointing this out. We agree with this comment. Therefore, we have added the total evolution time of DD gate: single-qubit gate ( Ry(π/2): 0.103 ms、 Rx(π/4): 0.0526 ms) and two-qubit gate (CRy(π): 0.169 ms). The duration of quasi-static noise can cover the entire gates operation time. The modified parts can be found in section 3.1 (page 6) and section 3.2 (page 7) (marked in red).
|
||
Comments 2: Although this paper is based on theoretical simulation, it would be more valuable if the feasibility of implementing DD gates in Si:P devices could be briefly discussed. What challenges will be faced in achieving the required DD pulse sequences and pulse control accuracy in the experiment. |
||
Response 2: Agree. Thank you for pointing this out. We have added relevant content about the challenges that will be faced in implementing the DD gates in the experiment, mainly including deviation of the controlled rotation angle within , system temperature rise caused by gate control, and system instability caused by heating, etc. References ([57]- [61]) have been added to support the claim. The modified parts can be found in the second paragraph of DISCUSSION AND CONCLUSION (page 12) (marked in red).
Comments 3: It is assumed in paper that the pulse is ideal. In fact, when applying the pulse sequences, errors will be introduced: the rotation angle may be too large or too small, and the rotation axis may also be offset. These are all important factors that bring errors. For these non-ideal cases, how effective is the DD gate sequences in enhancing the robustness of the system. Response 3: Agree. Thank you for pointing this out. We have added the figures of the variation of the fidelity of single-qubit gate (Ry(π/2+θerror) ) and two-qubit gate ( CRy(π+θerror)) control with the rotation angle error under the condition of quasi-static noise intensity of 1000 Hz (Figure B1). Taking the rotation angle error ( θerror) range of -π/8~π/8 as an example for illustration. The results show that, whether it is single-qubit gate or two-qubit gate control, the DD gate sequences have better anti-noise performance compared with the NMR pulse sequence and has a significant effect in enhancing the robustness of the system. The modified parts can be found in Appendix B (page 13) (marked in red).
Comments 4: This manuscript focuses on single-qubit and two-qubit operations. Could the authors discuss whether this approach has the same effect on multi-qubit systems? Response 4: Thank you for pointing this out. In principle,DD gate can be extended to the case of multi-donor system, and will be investigated in more details in the future. However, this content is beyond the scope of this paper's research. We will further expand on this issue in the future. The modified parts can be found in the third paragraph of DISCUSSION AND CONCLUSION (page 12) (marked in red).
Comments 5: The authors compared their method with traditional NMR. It would be more useful to briefly compare DD gates with high-fidelity control techniques such as GRAPE and DRAG. Response 5: Agree. Thank you for pointing this out. We have added a comparative discussion of DD gate with GRAPE and DRAG. This paper adopts the analytical method, while GRAPE relies on the numerical method. The waveforms are relatively complex, and it is difficult to generate complex waveforms under the condition of short electron coherence time. DRAG mainly optimizes for high-level leakage. Through discussion, the applicable scope and differences of each high-fidelity control technology were further clarified. References ([55]、 [56]) have been added to support the claim. The modified parts can be found in the second paragraph of DISCUSSION AND CONCLUSION (page 11) (marked in red).
Comments 6: If this paper involves specific software packages or numerical solvers, please supplement the references of the simulation tools or methods used. Response 6: Thank you for pointing this out. This article does not involve specific software packages and numerical solvers. We simulate it by solving the time evolution of the Schrodinger equation. Specifically, by discretizing the time and solving the differential equation system corresponding to the Schrodinger equation, the quantum state evolution under the time-dependent Hamiltonian is calculated for each time t. The evolved quantum state is obtained through numerical solutions, and finally the final state is obtained, and the fidelity of the quantum state is calculated. At the same time, the influence of noise can be simulated by introducing the environmental coupling term. |
The modified parts can be found in the last paragraph of Section 2.1(marked in red).
In addition, we have made certain modifications to the title of the article to more accurately reflect the core content and the key points after the revision.

Reviewer 2 Report
Comments and Suggestions for Authors
In the present manuscript, the authors provided an innovative scheme for implementing high fidelity quantum gate operations and Bell state preparation in a silicon-based phosphorus-doped (Si:P) system by utilizing dynamical decoupling gates (DD gates), thereby resolving the conflict between decoherence protection and qubit manipulation. The study demonstrates clear theoretical value and experimental guidance, with comprehensive data and a logically sound structure. Therefore, I recommend its publication after revising some necessary issues. The detailed questions are given as follows.
- Some terminology usage was found to be inconsistent in the present manuscript. For instance,the phrase "Dynamical Decoupling" is sometimes abbreviated as "DD" and at other times is written out in full, which lacks uniformity. Some content in the present manuscript appears redundant. For instance, the descriptions of DD gates appear multiple times in the Sections of Abstract and Introduction, leading to redundancy.
- Some derivations in the manuscript are based on assumptions (e.g., "assuming that each evolution time interval is sufficiently short"), but the authors do not verify thescope of applicability for these assumptions.
- In Sec. 2.2, i.e., the "System Hamiltonian" part, the authors give the mathematical expression of the Hamiltonian but do not give the explanation of its physical significance, which potentially makes it difficult for non-specialist readers to understand the complete physical model. There is a missing matrix element in Eq. (2). I recommend the authors further check the possible typographical or formatting errors.
- In the manuscript, the authors implement a universal set of high-fidelity quantum gates and prepare Bell states using dynamical decoupling gates (DD gates). Could the authors clarify the applicable regime of the DD gates? Are there any limitations to this approach? When discussing the advantages of DD gates, the authors only give the fidelity, but there is insufficient explanation for why DD gates perform better under noise. The authors should further give the theoretical support.
- In the present manuscript, the authors mention simulation results and list the experimental data but do not specify the detailed settings of the simulation parameter (e.g., the specific implementation of the noise model, the selection of the initial state, etc.), whichmake it lacks reproducibility.
- The captions for some figures and tables (e.g., Fig. 1, Fig. 2) are not sufficiently detailed, lacking explanations for key parameters in the figures/tables, which hinders the comprehension for readers.
- Table 1compares the gate fidelity of different quantum control schemes and highlights the advantages of the proposed method. However, the experimental conditions in the referenced works, such as the noise strength, are not discussed. It would be helpful to clarify whether these conditions are comparable to those used in this study.
- The formatting of references is inconsistent in the present manuscript. For instance, some author names are abbreviated, some are full, and some references lack page numbers or journal volume numbers. The authors should further check these formatting issues.
Round 2
Reviewer 1 Report
Comments and Suggestions for Authors
I have found that the authors have satisfactorily addressed my concerns. I would like to recommend publication as is.
Reviewer 2 Report
Comments and Suggestions for Authors
In the revised manuscript, the authors have substantially addressed the comments I had pointed out before. Overall, the description of the physical model and the discussion of the results become more clear than the previous version. However, there are still some minor details that require further revision.
- In the Abstract: “Dynamic decoupling(DD) can suppress decoherence caused by environmental noise, while in hybrid system it also hinder coherent manipulation...”“hinder” should be corrected as “hinders”.
- Onpage 1, In the last sentence, the the phrase “Dynamic decoupling” is missing.
- In Sec. 3.1: “The implementation process of the single-qubit gate is as following.” “as following” should be corrected as “as follows”.
- “The results show that the DD gates can effectively suppress the effect of environment noise...” “environment” should be corrected as “environmental”.